# Murine Typhus: A Review of a Reemerging Flea-Borne Rickettsiosis with Potential for Neurologic Manifestations and Sequalae

Lucas S. Blanton

Department Internal Medicine, Division of Infectious Diseases, University of Texas Medical Branch, Galveston, TX 77555, USA; lsblanto@utmb.edu; Tel.: +1-(409)-370-9759 or +1-(409)-772-6527

**Abstract:** Murine typhus is an acute febrile illness caused by *Rickettsia typhi*, an obligately intracellular Gram-negative coccobacillus. Rats (*Rattus species*) and their fleas (*Xenopsylla cheopis*) serve as the reservoir and vector of *R. typhi*, respectively. Humans become infected when *R. typhi*-infected flea feces are rubbed into flea bite wounds or onto mucous membranes. The disease is endemic throughout much of the world, especially in tropical and subtropical seaboard regions where rats are common. Murine typhus is reemerging as an important cause of febrile illness in Texas and Southern California, where an alternate transmission cycle likely involves opossums (*Didelphis virginiana*) and cat fleas (*Ctenocephalides felis*). Although primarily an undifferentiated febrile illness, a range of neurologic manifestations may occur, especially when treatment is delayed. Serology is the mainstay of diagnostic testing, but confirmation usually requires demonstrating seroconversion or a fourfold increase in antibody titer from acute- and convalescent-phase sera (antibodies are seldom detectable in the first week of illness). Thus, early empiric treatment with doxycycline, the drug of choice, is imperative. The purpose of this review is to highlight murine typhus as an important emerging and reemerging infectious disease, review its neurologic manifestations, and discuss areas in need of further study.

**Keywords:** murine typhus; endemic typhus; flea-borne typhus; encephalitis; meningitis; vasculitis; rats; opossums; fleas; *Rickettsia typhi*

## 1. Introduction

Murine typhus, also known as endemic typhus or flea-borne typhus, is an acute undifferentiated febrile illness [1]. The disease is caused by *Rickettsia typhi*, an obligately intracellular Gram-negative coccobacillus [2]. Murine typhus is endemic to much of the world, especially along seaboard regions of the tropics and subtropics, where rats and their fleas play a role in the maintenance and transmission to humans [3]. It is an underappreciated cause of febrile illness in many parts of the world, as confirmatory diagnosis is difficult, and signs/symptoms mimic various other infectious diseases [4]. In parts of the United States, the disease is reemerging as an important cause of febrile illness, especially in Southern California and throughout Texas. In these regions, an alternate cycle of transmission involving opossums and cat fleas is thought to drive the reemergence of murine typhus [5]. The ubiquity and cosmopolitan nature of the cat flea, the wide distribution of opossums in North America, and the apparent increase in incidence in the last two decades all sound the alarm that murine typhus has the potential to emerge in locations where it is not yet endemic. Although generally considered mild in severity, especially when compared to other rickettsioses such as Rocky Mountain spotted fever (RMSF) and louse-borne epidemic typhus, severe manifestations and death can occur [6–9]. Murine typhus has been associated with a range of neurologic manifestations, and although those on the more severe spectrum are relatively uncommon, their relative frequency will increase as the disease continues to emerge. This general review of the literature aims

to highlight murine typhus as an important emerging and reemerging infectious disease, review its neurologic manifestations, and discuss potential areas of further study to mitigate spread and prevent sequelae of *R. typhi*.

## 2. Microbiology

The genus *Rickettsia* is divided into four phylogenetic groups, which include the typhus, spotted fever, transitional, and ancestral groups (Figure 1) [10,11]. The typhus and spotted fever groups comprise the majority of clinically relevant pathogenic rickettsiae. While the spotted fever group includes over 20 tick-borne species, the typhus group is made up of only two species: *Rickettsia prowazekii* and *R. typhi*, which are primarily louse- and flea-borne, respectively. *Rickettsia typhi* is the bacterium responsible for murine typhus. *Rickettsia felis* is another species that infects fleas, but it does not belong to the typhus group. Rather, it is in the transitional group and is purported to cause febrile illness (flea-borne spotted fever) [10,12]. In addition to *R. felis*, other similar rickettsiae (referred to as *R. felis*-like organisms) have also been found within fleas [13,14].

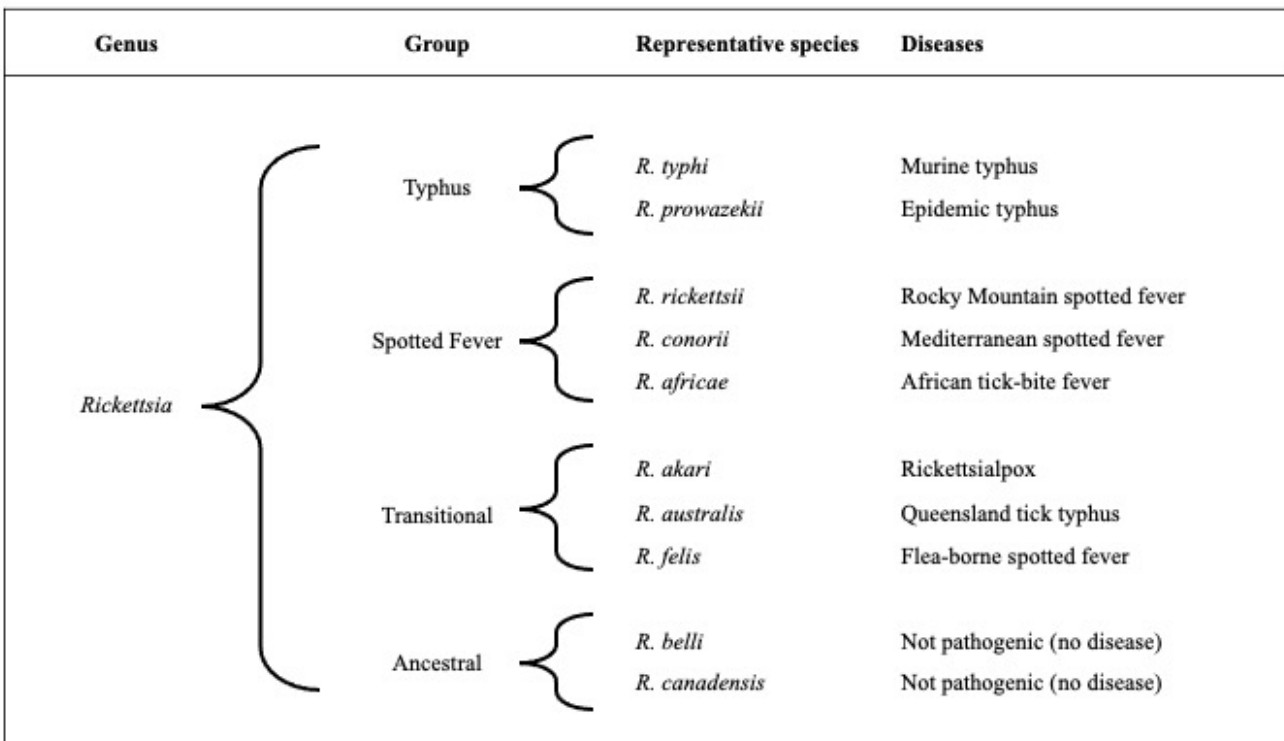

**Figure 1.** A schematic of the four rickettsial groups, representative rickettsial species, and their associated disease names.

*Rickettsia typhi* is a small 1.3 × 0.4 μm Gram-negative coccobacillus [2]. Through genome reduction, rickettsiae have evolved to develop a niche for obligate intracellular life using the host cell to provide necessary elements for survival—they are unable to produce the enzymes necessary for the synthesis of lipids, nucleotides, and the metabolism of carbohydrates [15]. *Rickettsia typhi* has a small 1.1 Mb genome [16], and isolates originating from distant geographic regions are genetically conserved [17], likely a result of dispersal of *Rattus* and rat fleas by trade involving sailing ship. The organism contains and expresses genes for several autotransporters (*sca1, sca2, sca3, sca4,* and *sca5*) [18]. The *sca5* gene encodes outer membrane protein B, which plays a role in cellular adhesion and entry [19,20]. Unlike organisms of the spotted fever group, *R. typhi* does not contain a functional gene for outer membrane protein A [20]. *Rickettsia typhi* encodes for numerous other genes that potentially play a role in its survival within the host [16]. Genes for a type IV secretion system and potentially membranolytic proteins could aid cellular invasion and escape from

the endosome [21–23]. A phosphatidylinositol 3-kinase effector is involved in autophagy to allow the organism to avoid autolysosomal destruction [24]. The presence of genes encoding numerous secretion system components may offer *R. typhi* methods to influence cell function [25,26]. Intracellular replication is facilitated by eluding IL-1α-mediated inflammasome induction [27].

In fleas, *R. typhi* infects the midgut epithelial cells [28]. In humans, the organism primarily infects endothelial cells [29]. Endothelial injury is the central pathophysiologic process that leads to the manifestations of murine typhus and in severe cases leads to neurologic sequelae and other end-organ manifestations (see sections below).

### 3. Pathology, Pathogenesis, and Immunity

Few autopsies of individuals who died as a result of murine typhus are detailed in the literature [30–33]. As with other severe rickettsioses, pathologic findings demonstrate systemic endothelial infection. The characteristic lymphohistiocytic vasculitis may be found in any organ and includes interstitial pneumonitis, portal triaditis, interstitial nephritis, and myocarditis. In regard to neuropathology, detailed descriptions are only noted in four autopsy cases [30–32]. Gross changes of the central nervous system, reported in three of these cases, were described as cerebral edema and congestion, varying in severity from mild to severe [31,32]. In one case, mild cerebellar tonsilar herniation was noted [31]. Microscopically, findings in the brain have included multifocal petechiae in the white matter, ischemic necrosis of neurons within the cerebral and hippocampal watershed areas, and perivascular mononuclear infiltrates—the typical vascular injury associated with rickettsial infection [30–32].

The typhus nodule—a classic finding in those with louse-borne epidemic typhus—has been noted, albeit not universally, in some fatal cases of murine typhus [31,34]. Typhus nodules represent inflammatory nodule-like lesions located within the central nervous system. They are composed of lymphocytes, plasma cells, and glial cells located around blood vessels. Typhus nodules are a well-described neuropathological finding in those with louse-borne epidemic typhus and were once thought to be relatively specific for *R. prowazekii* infection. The paucity of histopathologic descriptions of fatal murine typhus, compared to the well documented features of the much more severe louse-borne typhus, obscures the true prevalence of typhus nodules in those with murine typhus. A retrospective examination of seven archived formalin-fixed, paraffin-embedded tissue blocks (originating from patients believed to have died of louse-borne typhus in Hamburg, Germany during WWII) amplified *R. typhi* DNA from blocks of two of these historical cases. These cases were also noted to have typhus nodules. Immunohistochemical studies on these tissues demonstrated that lymphocytes, the predominant cellular infiltrate, were composed of CD8 and CD4 T cells in a roughly 60% to 40% mix, respectively. Microglia and macrophages were also identified by immunohistochemical staining, but B cells and neutrophils were rarely present. In addition to showing that *R. typhi* played a role as a cause of death in war-ravaged Europe, the study also demonstrates that typhus nodules occur as a result of infection with both *R. prowazekii* and *R. typhi* [34].

Other neuropathological findings have included mononuclear infiltration of the meninges in two autopsy cases [32] and perivasculitis of the pituitary in two others [30,32]. One report described the involvement of the spinal cord, which consisted of perivascular infiltrates of macrophages and lymphocytes [30].

*Rickettsia*-induced endothelial damage is the hallmark pathophysiologic process that leads to the severe manifestations of murine typhus and other rickettsioses [29]. As injury accumulates systemically, extravasation of intravascular fluid into the interstitium can occur, contributing to hypovolemia and decreased organ perfusion. This can lead to acute kidney injury as a result of prerenal azotemia and eventual acute tubular necrosis. In the lung and central nervous system, endothelial damage can lead to pneumonitis and meningoencephalitis, respectively. Hepatic injury, a frequent occurrence manifesting as elevated hepatic transaminases on routine blood work, results from multifocal infection of

portal and hepatic sinusoidal endothelium. Occular manifestations of these processes are evident by choroidal vascular injury noted on retinal examination [35].

The primary mechanism of *R. typhi* inoculation in humans is rubbing or scratching infected flea feces into flea bite wounds, abrasions, or onto mucous membranes. Considering the cutaneous route of inoculation, as with other organisms in the genus *Rickettsia*, dendritic cells are a crucial element of the early immune response [29]. Natural killer cells are an important aspect of controlling rickettsiae during early infection. Adaptive responses with CD8 and CD4 T lymphocytes follow [36–39].

## 4. Ecology of an Emerging and Reemerging Infection

In the early part of the 20th century, in North America, it was recognized that there existed an illness that resembled louse-borne epidemic typhus, but unlike the classic form, it was milder and did not occur in large outbreaks [40–42]. The lack of household clustering, occurrence during warmer weather, lack of associated body louse infestations, and the apparent link to food stuffs led to the hypothesis that the disease centered around rodents and their ectoparasites [43]. Indeed, Dyer and Mooser would later independently confirm that rats and their fleas harbored bacteria responsible for this endemic form of typhus [44–46]. Throughout most of the world, rat species (*Rattus rattus* and *R. norvegicus*) and rat fleas (*Xenopsylla cheopis*) are the reservoirs and vectors of *R. typhi*, respectively (Figure 2) [47]. Rats are not grossly affected by infection with *R. typhi*, but prolonged bacteremia allows *X. cheopis* to acquire infection during feeding. Transmission is primarily horizontal. In fleas, the bacterium establishes itself within midgut epithelial cells, which are then shed into the feces [48]. The organism is also vertically passaged to flea progeny [49]. Infection does not affect flea lifespan or fecundity [50]. Murine typhus is epidemiologically linked to rats throughout much of the world and was closely linked to cases in the U.S. prior to intense vector control efforts beginning in the mid-1940s [3,51]. The ubiquity of rats throughout the world and their ability to thrive where humans are established make the possibility of emergence in non-endemic areas possible [47].

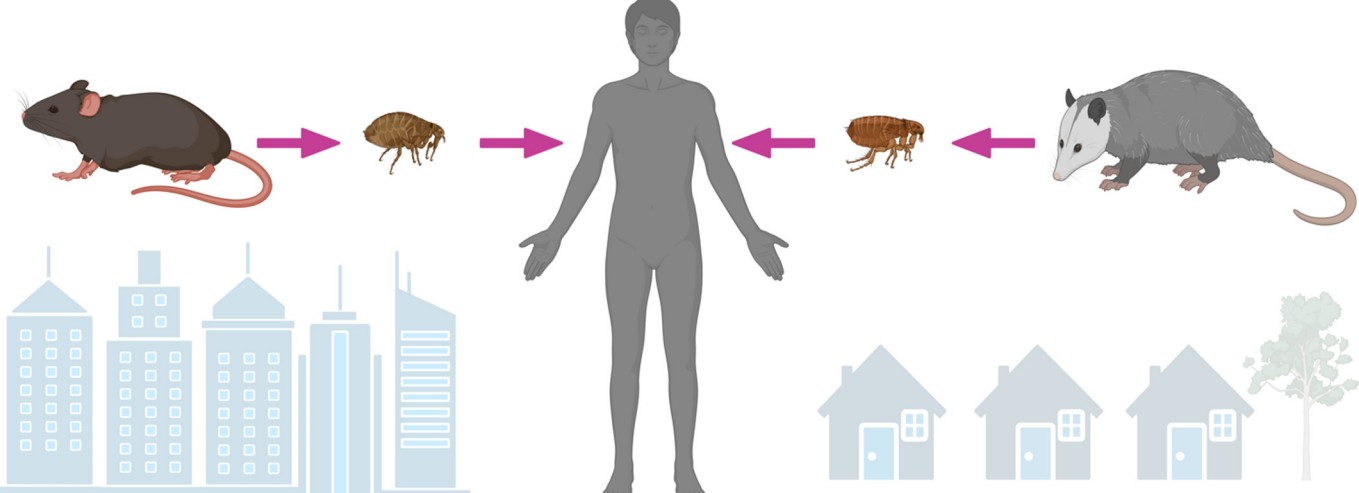

**Figure 2.** *Rickettsia typhi* is maintained by *Rattus* species and transmitted to humans by rat fleas (*Xenopsylla cheopis*), especially in the urban setting. *Rickettsia typhi* likely utilizes opossums (*Didelphis virginina*) as an amplyfying host in suburban areas of North America, where it is transmitted to humans by cat fleas (*Ctenocephalides felis*).

During World War II, dichlorodiphenyltrichloroethane (DDT) was used effectively to prevent malaria and outbreaks of louse-borne epidemic typhus. After the war, it became commercially available in the U.S. and was widely implemented in various fashions to control insect pests for home and agricultural purposes [52]. Although rat control measures had been previously implemented in campaigns to control murine typhus, it was not until

DDT was systematically used on rat runs and rat harborages that traction was achieved in the control of murine typhus [53,54]. Approximately 5400 cases were reported at the U.S.'s peak incidence in 1944. By 1956, after the initiation of targeted DDT dusting, reported cases of murine typhus fell to less than 100 [51]. The effectiveness of DDT was noted by decreasing rat flea infestation rates [55] and from the fall of cases in counties targeting rats with DDT versus counties not participating in this vector-control measure [54].

Interestingly, despite the apparent disappearance of cases in much of the U.S., small numbers of cases continued to be reported in parts of Southern California and the southernmost counties of Texas [56,57]. In these areas, an alternate cycle of transmission involving opossums (*Didelphis virginiana*) and cat fleas (*Ctenocephalides felis*) exists, as first established in an ecologic study performed in suburban areas of Los Angeles, California (Figure 2) [57]. Most evidence supporting *D. virginiana* as a link to cases of murine typhus has been circumstantial (opossums have a high seroprevalence to typhus group rickettsiae and harbor *R. typhi*-infected cat fleas) [57–61]. More direct evidence—isolation of the bacterium from the spleen and prolonged rickettsemia after experimental infection—has been demonstrated from only a single opossum [57,62]. In an animal model, using *Monodelphis domestica* (the laboratory opossum) as a surrogate to *D. virginiana*, intradermal inoculation of *R. typhi* caused prolonged rickettsemia, infected tissues, and pathology typical of a rickettsiosis. Despite disseminated infection, these opossums remained clinically well, suggesting their ability to act as an amplifying host [63].

Experimentally, the cat flea (*C. felis*) has demonstrated the ability to acquire *R. typhi* [64]. This flea species has long been implicated as a possible vector for *R. typhi* [56,65,66]. In addition to frequently infecting opossums, *C. felis* infests domestic cats (as its name implies), dogs, and a variety of other mammals [67]. The role of domestic animals, such as cats, in the transmission cycle of *R. typhi* is not well established. Experimental infections have resulted in short durations of bacteremia [68], and the low prevalence of *R. typhi*-infected fleas collected from cats do not support their role as being a principal element to the bacterium's maintenance and transmission [69–71]. The cosmopolitan nature of the cat flea supports the possibility of *R. typhi* emerging in non-endemic areas [67].

## 5. Epidemiology of an Underrecognized and Reemerging Cause of Infection

Murine typhus is a cause of undifferentiated febrile illness throughout much of the world [3]. It is most recognized in tropical and subtropical regions, especially port cities where rats, the primary mammalian reservoir, thrive [72]. It is believed that for every case of murine typhus that achieves a diagnosis, several others remain unrecognized [47]. Indeed, there is a surprising seroprevalence rate in regions where the disease is reemerging or is not considered highly endemic. This indicates that many cases remain undiagnosed or are not reported [73–76]. Cases increase in frequency in the spring, peak in the summer, and decrease in the fall [3,77,78].

Outbreaks of murine typhus occur when there exists inadequate reservoir and vector control [79,80]. The disease is well recognized in Southeast Asia, where it is an important cause of fever (especially in urban areas) in Thailand [81], Laos [82,83], Indonesia [84], and Vietnam [85]. There are also contemporary reports of murine typhus in Asia, where it has been reported in Nepal [86], Bangladesh [87], and India [88]. Murine typhus is endemic to southern European and other countries around the Mediterranean [9,89,90]. Once a frequently reported disease in South America, murine typhus is now seldom recognized as a cause of febrile illness. It is unclear if there was a true decline, perhaps related to the broad use of DDT in malaria control programs, underreporting, or lack of clinical recognition. When sought by clinicians and researchers, the presence of murine typhus as a reemerging disease entity in South America becomes evident [91]. In North America, cases are recognized in Mexico [92,93] and in the U.S. states of Hawaii, California, and Texas [94–96]. There has been an alarming increase in California, with foci around homeless encampments, and in Texas, where the distribution of disease is moving northward [33,96,97]. The increased recognition of cases in Southern California, Texas, Mexico, and South America

highlights the reemergence of the pathogen in these regions. It has also been described in travelers where it has been reported to have been acquired via travel to Asia, Africa, and the Americas [98–102].

### 6. Clinical Manifestations of an Acute Undifferentiated Febrile Illness

Murine typhus is an undifferentiated febrile illness with signs and symptoms that mimic a variety of infectious and noninfectious syndromes (see the diagnosis section below for the differential diagnosis). The illness usually starts abruptly [103] after an incubation period of 4 to 15 days [104]. Early symptoms accompanying fever include headache (detailed in the neurologic manifestations section below), chills (63%), malaise (67%), and myalgias (52%). Gastrointestinal symptoms vary in frequency and may include anorexia (48%), nausea/vomiting (27%), diarrhea (19%), and abdominal pain (18%). Cough occurs in 27% and develops as the illness progresses. Less common symptoms include conjunctivitis (18%), sore throat (14%), and photophobia (10%) [8].

An important sign (also sometimes reported as a symptom) of murine typhus is rash. Although often considered a sine qua non of rickettsial illness, rash occurs in only about half of those with murine typhus. It is seen in 18% at presentation [6] and occurs in 48% at some point during illness [8]. The rash is described as macular (49%), maculopapular (29%), papular (14%), and seldom petechial (6%). It is most often distributed on the trunk (88%), but also occurs on the legs (45%), arms (37%), hands (5%), and feet (5%) [6]. The palms and soles are involved in less than 3% [78]. When present, the rash can be subtle, and it is difficult to detect in those with darkly pigmented skin. Thus, it is more frequently described in those with lightly pigmented skin (81%) than those with darkly pigmented skin (20%) [104]. Other physical signs include hepatomegaly (22%), splenomegaly (17%), and lymphadenopathy (13%) [8].

A variety of laboratory abnormalities accompany the illness. None of these are specific for murine typhus and can be seen in a variety of other febrile illnesses. Abnormalities on a complete blood count may reveal thrombocytopenia (42%), anemia (38%), leukopenia (24%), and leukocytosis (18%) [8]. Abnormalities on metabolic panels (indicative of systemic endothelial damage and extravasation of intravascular fluid into the interstitium) include hypoalbuminemia (60%), hypoproteinemia (45%), and hyponatremia (35%) [6,8]. The latter is a result of the appropriate secretion of antidiuretic hormone [105]. Hepatic and cellular injury are manifested with serum elevations in hepatic transaminases (79%), lactate dehydrogenase (73%), and alkaline phosphatase (41%). Creatinine kinase levels are elevated in 29% [8]. Extremely high levels of creatine kinase, as a result of frank rhabdomyolysis, have been reported to occur during the course of murine typhus [106]. C-reactive protein and procalcitonin are often elevated [107].

Most patients with murine typhus will recover uneventfully, but illness can be protracted, with fever and other symptoms lasting upward of 3 weeks. Severe sequelae of infection can occur, however. Acute kidney injury is a result of prerenal azotemia. When severe or prolonged, acute tubular necrosis can result [108–110]. Hemodialysis is sometimes temporarily needed. Respiratory failure requiring mechanical ventilation can also occur [111]. Severe rickettsial disease has been noted to be associated with the following risk factors: alcoholism, glucose-6-phosphate deficiency, and the use of sulfonamide antibiotics [112,113]. The case fatality rate of murine typhus is 0.4% in both the pre- and post-antibiotic eras [7,114]. The similar contemporary case-fatality rate likely indicates a subset of individuals where either effective treatment was not received or it was administered late during the course of disease. Of those hospitalized, 10% require intensive care, and the case-fatality is as high as 4% [6].

### 7. Neurologic Manifestations and Sequelae

The most common neurologic manifestation of murine typhus is headache. Occurring in 81%, it is the most frequent complaint after fever [8]. It is usually frontal in location, but it is sometimes distributed occipitally [103]. Headache manifests early and persists through-

out the febrile course; it subsides during defervescence. It is often described as the worst symptom associated with murine typhus, and there is little relief with analgesics [103,104]. Other less frequent neurologic manifestations include confusion or delirium (8%), stupor (4%), seizures (4%), and ataxia (1%) [6].

The prominence of fever, severe headache, and the occasional complaints of photophobia (10%) [8] and nuchal rigidity (6%) [104], may warrant obtaining a lumbar puncture for analysis of cerebrospinal fluid (CSF). In two early publications detailing cases of murine typhus, symptoms prompted lumbar puncture in 20 (11%) and 45 (36%) of 120 and 126 cases, respectively [103,104]. When obtained, the CSF is most often normal. When indicative of meningoencephalitis, the CSF resembles that of many viral etiologies, with a clear appearance and low white cell count compared to conventional community-acquired bacterial causes (median white cell count of 10 vs. 410, respectively). Elevated protein concentration is noted in 46% [115]. Hypoglycorrhachia has been reported with CSF to blood glucose ratios < 0.5 in 38% [115,116]. Unfortunately, there are no specific biomarkers to differentiate *R. typhi*-induced central nervous system injury from that of other infections [117]. Meningitis or meningoencephalitis has been described to occur in approximately 2% of individuals with murine typhus [118], but in a large case series of patients from Canary Islands, Spain, a series reporting a relatively high rate of severe manifestations, meningitis occurred in 5.6% [9]. As in other rickettsial diseases, the case fatality rate of murine typhus is quite high—27%—when meningoencephalitis occurs [115].

Other seemingly rare but severe neurologic manifestations have been reported: cranial nerve palsies (facial and abducens) [119–121] and status epilepticus [31,122]. Most patients recover with no neurologic sequelae. This is unlike what has been described for more severe rickettsioses, such as RMSF [123–125]. Nevertheless, long-term neurocognitive disability has been reported in a few patients diagnosed with murine typhus [126–128].

Ocular manifestations occur but are often subclinical. In a small series of patients receiving formal ophthalmologic exams during the course of murine typhus, eight of nine (89%) had bilateral ocular involvement attributed to the infection, but most of these patients (63%) had no ocular symptoms [35]. When symptoms occur, complaints include decreased visual acuity, blurry vision, floaters, ocular redness, and conjunctivitis [35,129]. Sudden unilateral loss of vision due to optic neuritis has been reported [35]. White retinal lesions and retinal hemorrhages are noted most often on exam, but optic disk edema has also been noted. With treatment, ocular involvement is self-limited, as demonstrated by resolving fundoscopic changes on follow up exams [35,129]. Parinaud's oculoglandular syndrome (unilateral conjunctivitis with associated ipsilateral regional lymphadenopathy) as a result of murine typhus has been reported [130,131].

## 8. Diagnosis

The early diagnosis of murine typhus relies on a high index of clinical suspicion. Early recognition that a febrile illness may be due to a *Rickettsia* species is paramount, as empiric treatment to avoid severe manifestations or death, is necessary while awaiting confirmatory diagnostic testing. As an undifferentiated febrile illness, the differential diagnosis is extensive. Other infections caused by organisms in the family *Rickettsiaceae* (i.e., spotted fever group rickettsioses, rickettsialpox, louse-borne typhus, flying-squirrel-associated typhus, and scrub typhus) and *Anaplasmataceae* (e.g., ehrlichioses and anaplasmosis) present similarly. Endocarditis, meningococcemia, disseminated gonococcal infection, secondary syphilis, relapsing fever, leptospirosis, typhoid fever, rubella, rubeola, roseola, mononucleosis from Ebstein–Barr virus and cytomegalovirus, acute retroviral syndrome, and arboviral infections such as dengue fever all have overlapping signs and symptoms. Finally, a variety of noninfectious syndromes (e.g., thrombotic thrombocytopenic purpura, immune thrombocytopenic purpura, Kawasaki disease, and various vasculitides) should be considered [132].

Serology is the primary diagnostic method to confirm murine typhus, with the indirect fluorescent antibody (IFA) assay being the mainstay serologic test [133]. Antibodies are

seldom present in the first few days of illness though. Roughly 50% of those with murine typhus will have detectable antibodies within a week of illness onset. By the second week of illness, diagnostic titers are present in almost all patients [6]. Confirmatory serologic diagnosis requires seroconversion or fourfold rise in titers from testing performed on acute- and convalescent-phase specimens [134]. Unfortunately, the IgM isoform does not appear much earlier than IgG and may suffer from more cross-reactivity. A single reactive IgG during a clinically compatible illness may be supportive for the diagnosis of murine typhus, but it must be noted that anti-*R. typhi* antibodies can persist for some time (the median titer a year after infection is 1:800) [135]. Thus, the presence of reactive antibodies may reflect a previous illness. In Galveston County, Texas, and on Honshu Island, Japan, seroprevalence studies have demonstrated a seroprevalence of 7.8% and 7.7% at titers of at $\geq$1:128 and $\geq$1:160, respectively [76,97]. Thus, in areas where the disease is endemic, there may be a substantial seroprevalence within the population.

Enzyme-linked immunosorbent assays for diagnosing typhus group rickettsioses have been developed, but they are not yet in regular use and still require acute- and convalescent-phase specimens [136]. The Weil–Felix reaction has proven to be both insensitive and nonspecific for the diagnosis of rickettsioses [137]. *Rickettsia typhi* and *R. prowazekii* share similar antigens, so antibodies stimulated by infection with *R. prowazekii* will react to diagnostic antigens derived from *R. typhi*. Thus, a species-specific diagnosis is not possible from standard serologic assays. Cross-absorption techniques have been employed to determine species-specific reactivity, but this testing is cumbersome and only performed in the research setting [138].

Several nucleic acid amplification techniques have been studied for the detection of rickettsiae. These include conventional and quantitative real-time polymerase chain reaction (PCR), loop-mediated isothermal amplification (LAMP), and recombinase polymerase amplification (RPA) [139,140]. Despite the availability of platforms with excellent analytic sensitivity, such as quantitative real-time PCR (which can detect only a few copies of target DNA per reaction volume), the limited number of circulating organisms limits the utility of PCR to detect *R. typhi* from clinical blood specimens. The median clinical sensitivity of PCR from blood and skin specimens is 5% [134]. LAMP and RPA methods are attractive technologies that can be employed in the field or in rudimentary laboratory settings where a thermocycler may not be available. As with PCR, these techniques are likely limited by the number of circulating rickettsiae within the blood. Detection of 23S rRNA, a target that has multiple copies, via reverse transcriptase PCR may offer enhanced analytic sensitivity compared to PCR assays targeting single-copy genes [141]. Next-generation sequencing has been used to amplify *R. typhi* DNA from clinical blood specimens [142–144].

The direct detection of *R. typhi* can be accomplished by immunohistochemical techniques, using typhus group-specific antibody, to directly visualize the organism within formal fixed, paraffin-embedded tissue sections [30,145]. The clinical performance of immunohistochemistry (IHC) for diagnosing murine typhus is unknown, but the test has a 70% sensitivity and 100% specificity for RMSF [132]. The lack of rash for biopsy in approximately 50%, and the few laboratories performing IHC limit the utility of this technique to the average clinician.

The isolation of *R. typhi* in culture is seldom undertaken. It requires specialized cell culture techniques and should be performed in biosafety level 3 laboratory conditions to avoid aerosolization and infection of laboratory workers. Thus, isolation is generally only performed in the research setting.

## 9. Treatment and Prevention

It is important to start immediate empiric therapy when murine typhus is suspected, as current diagnostics fail to reliably offer confirmatory results in a timely manner. Prompt treatment can quickly abate symptoms, such as fever and headache, and prevent progression to severe manifestations [146], such as central nervous system complications. Length

of hospitalization has been demonstrated to be shorter when early suspicion leads to prompt treatment [147].

Tetracyclines are the drug class of choice, with doxycycline being the preferred agent [148]. The in vitro susceptibility testing of antibiotics against rickettsiae is not standardized nor available through commercial reference laboratories, but studies using methods employing cell culture and embryonated hens' eggs have demonstrated minimum inhibitory concentrations (MIC) of 0.06–0.25 µg/mL for tetracyclines [149]. The use of doxycycline is supported by a wealth of observational data documenting its successful use [150]. A randomized open-label controlled trial, comparing doxycycline versus azithromycin, found that patients defervesced at a median time of 34 h after starting their first dose of doxycycline [146]. For adults, doxycycline is given at 100 mg twice daily. Many clinicians give a one-time 200 mg loading dose, followed by 100 mg twice daily. Doxycycline is well absorbed, so oral administration is usually adequate. When nausea/vomiting or critical illness prevents reliable enteral absorption, the intravenous form should be used. A duration of 5 to 7 days is sufficient. Older tetracyclines are effective but must be taken more frequently and have more adverse events [151]. Newer tetracycline-like antibiotics (i.e., tigecycline, eravacycline, and omadacycline) have activity against *R. typhi* [152], but clinical experience with these drugs during rickettsial illness is limited [153–155].

A few other antibiotics have been found to inhibit the growth of rickettsiae in vitro. These include chloramphenicol, some macrolides (i.e., clarithromycin and azithromycin), fluoroquinolones, and rifampin. Chloramphenicol (50–75 mg/kg/day divided into four doses) has historically been used as an alternative to tetracyclines [150]. A case series evaluating the effectiveness of different regimens for murine typhus suggests it takes about a day longer to defervesce on chloramphenicol compared to doxycycline [156]. Chloramphenicol is no longer available in the U.S.

Murine typhus has been successfully treated with fluoroquinolones, such as ciprofloxacin. In a retrospective examination, the time to defervesce on ciprofloxacin was longer than doxycycline (4.2 days and 2.9 days, respectively) [156]. Treatment failures have been noted [150,157]. Azithromycin has been used, but it was deemed inferior to doxycycline in a controlled trial. There were more clinical failures (22.5% vs. 1.4%), and the mean time to defervesce was longer (48 h vs. 34 h) for azithromycin compared to doxycycline, respectively [146]. Many commonly prescribed antibiotics, such as trimethoprim-sulfamethoxazole, cephalosporins, and penicillins, have no in vitro activity against rickettsiae [149]. In fact, sulfonamide use is considered a risk factor for developing severe rickettsial illness [158].

Doxycyline (2.2 mg/kg twice daily) is the preferred treatment for children [159]. Short and infrequent courses of doxycycline do not cause noticeable color changes in developing permanent teeth [160–162]. The use of doxycycline for rickettsioses in young children is endorsed by the American Academy of Pediatrics Committee on Infectious Diseases [159]. Most pregnant women with murine typhus have favorable outcomes, but poor neonatal outcomes from those with suboptimal treatment have been reported [163,164]. Doxycycline does not seem to be associated with the severe adverse events in pregnant women that have been reported with older tetracyclines [165,166].

There is no available vaccine for the prevention of murine typhus or other rickettsioses. Identification of antigens that stimulate CD4+ and CD8+ T cells to secrete protective cytokines, the stimulation of cytotoxic T cells, and the stimulation of anti-rickettsial antibodies are all believed to be crucial in vaccine development. In experimental models, reactivity to rickettsial outer membrane protein B may offer protection from subsequent *R. typhi* infection [29].

The aforementioned campaign using DDT to control rat flea populations, as well as the subsequent fall of cases in the United States, exemplifies the effectiveness of vector control measures to curb disease transmission [51,54]. The extensive use of DDT in Latin America to control malaria is temporally associated with a marked decrease in reporting or recognition of murine typhus in this region [91,167]. It is unknown if similar methods to

control fleas on other mammalian hosts (i.e., opossums) would effectively control murine typhus. In one study, DDT used on rat runs and harborages failed to cause the collateral effect of controlling flea infestations on opossums trapped in areas where these treatments were applied [168]. Although there is no direct supportive evidence, it seems prudent to control fleas around homes, yards, and domestic animals (i.e., cats and dogs) to prevent the occurrence of murine typhus in endemic areas.

## 10. Conclusions and Future Directions

Murine typhus is an often-overlooked cause of febrile illness. Undifferentiated in regard to signs and symptoms, it resembles a variety of other infections. It is therefore difficult to recognize, often delaying necessary doxycycline treatment [169]. When treatment is delayed, prolonged illness, severe manifestations (e.g., respiratory failure, renal failure, meningoencephalitis), or death may occur. The reemergence of murine typhus in parts of California and Texas is an alert to other regions, as the reservoirs and vectors of *R. typhi* are widely distributed. As the prevalence of murine typhus increases, so will the number of cases complicated by life-threatening manifestations.

A key endeavor to curb disease progression from a mild febrile illness to one with severe life-threatening consequences is the development of rapid tests effective at early stages of illness. Doxycyline is a very effective therapy, but a rickettsiosis must be clinically recognized to prompt empiric use. The ability to offer clinicians sensitive and specific results at the point-of-care would provide the necessary diagnostic data to guide accurate medical decision making. An ideal test might include the detection of a secreted serum biomarker via an easy-to-use lateral flow assay. Developing targeted methods to break the transmission cycle to humans is also paramount. The dramatic post-World War II decline in the incidence of murine typhus throughout the U.S. is an excellent example of how vector-control efforts were able to limit disease transmission. Unfortunately, the strategy of using DDT had collateral environmental impacts. Novel targeted strategies to control fleas on reservoir hosts (i.e., opossums) in endemic areas may offer an alternative means to breaking the cycle of transmission. Such methods might include the use of baits, targeting opossums, laden with commercially available orally ingested anti-flea medications (e.g., spinosad).

**Funding:** This research received no external funding.

**Institutional Review Board Statement:** Not applicable as this work does did not involve human subjects.

**Informed Consent Statement:** Not applicable as this work does did not involve human subjects.

**Data Availability Statement:** No data sets were generated by the author for the purposes of this paper. Data presented in the context of this review have been previously published by others and cited within the manuscript.

**Acknowledgments:** The author would like to thank David H. Walker for his thoughtful review of this manuscript and his continued mentorship.

**Conflicts of Interest:** The author declares no conflict of interest.

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
