# Peer review of "Murine Typhus: A Review of a Reemerging Flea-Borne Rickettsiosis with Potential for Neurologic Manifestations and Sequalae"

_2036-7449, doi:10.3390/idr15060063_

Round 1

Reviewer 1 Report

Comments and Suggestions for Authors

In the article, "Murine Typhus: A Review of a Reemerging Flea-Borne Rickettsiosis with Potential for Neurologic Manifestations and Sequelae", the author records the objective of reviewing murine typhys an important emerging and re-emerging infectious disease, addressing neurological manifestations and discussing areas that require more in-depth studies. 

I did not understand that the author discuss areas of additional study to mitigate the spread and precente sequelae, but they discuss diagnostic methods and techniques in studies raised int the review that confirm the presence of the Rickettsia bacteria and the presence of antibodies against this bacteria. 

They also highligth in the treatment and prevention section that current diagnostic results are also unable to offer confirmatory results and hence the importance of initiating immediate empirical therapy.

The author carried out a very extensive bibliographical survey totaling 169 references. I got the impression that the author covered more topics about murine typhus than the title and objective indicated. The scientific literature on the subject was reviewed from 1911 (date of the firs published study related to the topic of this article until 2023). 

The author does not explain which databases he used for this survey. I observed that there were no inclusion or exclusion criteria for references. Were all articles suitables for what the author set out to answer? 

It wolud be interesting to follow the recommendations of the Preferred Peporting Items for Systematic Reviews and Meta-Analyses (PRISMA) guide for a sistematic literature review .

There are the observations I have for the moment. 

Author Response

Reviewer 1:

Author response:  Thank you for your thoughtful review.  I believe the suggestions made by yourself and the other reviewers have resulted in changes that greatly improve the manuscript. I have responded to comments below and have included approximate line numbers indicating changes to the manuscript.  I have also highlighted changes/additions in the revised manuscript. 

In the article, "Murine Typhus: A Review of a Reemerging Flea-Borne Rickettsiosis with Potential for Neurologic Manifestations and Sequelae", the author records the objective of reviewing murine typhys an important emerging and re-emerging infectious disease, addressing neurological manifestations and discussing areas that require more in-depth studies. 

I did not understand that the author discuss areas of additional study to mitigate the spread and precente sequelae, but they discuss diagnostic methods and techniques in studies raised int the review that confirm the presence of the Rickettsia bacteria and the presence of antibodies against this bacteria. 

Author response:  In regard to mitigating spread and preventing sequelae of infection, the last paragraph addresses this as future areas of study.  I have modified line 456-457 to further explain possible methods to break the transmission cycle to humans.

They also highligth in the treatment and prevention section that current diagnostic results are also unable to offer confirmatory results and hence the importance of initiating immediate empirical therapy.

The author carried out a very extensive bibliographical survey totaling 169 references. I got the impression that the author covered more topics about murine typhus than the title and objective indicated. The scientific literature on the subject was reviewed from 1911 (date of the firs published study related to the topic of this article until 2023). 

The author does not explain which databases he used for this survey. I observed that there were no inclusion or exclusion criteria for references. Were all articles suitables for what the author set out to answer? 

Author response:  The submitted manuscript is intended to be a review, but it is not intended to be a structured systematic review or meta-analysis.  This is why the manuscript is structured, as the Infectious Disease Reports author guidelines instruct, with front matter, literature review sections, and back matter.  The journal guidelines state that a systematic review be structured as a research manuscript with materials and methods, results, and discussion sections.  As a general review (not a “systematic review” or meta-analysis) this author is very hesitant to structure it in any way that may give the reader a false impression.  Therefore, structure of the PRISMA guidelines for conducting a systematic review or meta-analysis do not apply.  The citations in this manuscript are a collection of many years of studying, reviewing, and conducting research on rickettsioses.  References have been obtained through PubMed and cross referencing other published work while focusing research in the area of rickettsioses for over a decade. The sentence at line 47-49 has been revised to highlight that this is a general literature review.  I also performed a search of the manuscript for the term “systematic” or “meta-analysis.”  Besides when citing published works, I confirm that these terms are not used to describe the type of manuscript intended for this submission.    

It wolud be interesting to follow the recommendations of the Preferred Peporting Items for Systematic Reviews and Meta-Analyses (PRISMA) guide for a sistematic literature review .

Author response:  Please see response to the above query addressing the use (or in this case absence) of PRISMA methods.    

There are the observations I have for the moment. 

Reviewer 2 Report

Comments and Suggestions for Authors

In the review article titled, "Murine Typhus: A review of.... Sequalae", Dr. Blanton highlights murine Typhus as an important emerging and reemerging infectious disease. The review is thoroughly researched, well-written, and expertly compiled. Please see my suggestions to enhance this review's thoroughness, please see below:

1. Please format the names of bacteria in italics throughout the paper.

2. To enhance the presentation of certain information and to make it easier for the readers, please consider utilizing tables or figures. Given the volume of data, using tables or diagrams would aid in comprehension for the sections including microbiology, epidemiology, and clinical manifestations.

3. Please add reference for line 136

4. Line 143-144, please re-write this sentence in past tense.

5. Although there are no available vaccines, under Treatment and prevention section, please mention of any potential vaccine candidates or provide references pertaining to current research on the Murine Typhus vaccine.

6. Line 447, please elaborate what kind of novel targeted strategies in 1-2 sentences.

Author Response

Reviewer 2:

Author response:  Thank you for your time in putting together this review.  I believe the suggestions made, and the resultant changes, will greatly improve the manuscript.  I have addressed comments below and have included approximate line numbers indicating changes.  I have also highlighted changes/additions in the revised manuscript. 

In the review article titled, "Murine Typhus: A review of.... Sequalae", Dr. Blanton highlights murine Typhus as an important emerging and reemerging infectious disease. The review is thoroughly researched, well-written, and expertly compiled. Please see my suggestions to enhance this review's thoroughness, please see below:

  1. Please format the names of bacteria in italics throughout the paper.

Author response:  I apologize for this error.  All genus and species names were placed in italics when submitted.  The original manuscript was submitted simply as a double-spaced Word document.  It seems that the manuscript has since been placed into a templated document.  I believe some of the formatting was lost in this transition (at least in the first several paragraphs).  I have re-italicized the genus and species in these sections. 

  1. To enhance the presentation of certain information and to make it easier for the readers, please consider utilizing tables or figures. Given the volume of data, using tables or diagrams would aid in comprehension for the sections including microbiology, epidemiology, and clinical manifestations.

Author response:  The reviewer offers an excellent suggestion.  The following figures have been added to increase clarity and add visual appeal:

- Figure 1 (see line 842):  A general schematic of rickettsial groups, representative species, and diseases to enhance the microbiology section.

- Figure 2 (see line 849):  A figure summary of the transmission cycle of R. typhi to enhance the ecology section. 

  1. Please add reference for line 136

Author response:  There is no definitive reference for this statement.  It is the belief of some prominent rickettsiologists and based on what is currently known about the immunological response to rickettsial infection.  Thus this is expert opinion.  Because there is no clear reference to easily support this statement, this sentence has been removed. 

  1. Line 143-144, please re-write this sentence in past tense.

Author response:  The statements regarding Dyer and Mooser are in past tense.  The statement regarding rats as the reservoir and vector are not intended to be past tense, as rats remain the primary reservoir in most of the world.  I added a qualifier (“most of the world” on line 144) to help differentiate this cycle of transmission from the cycle occurring in California, Texas, and parts of Mexico. 

  1. Although there are no available vaccines, under Treatment and prevention section, please mention of any potential vaccine candidates or provide references pertaining to current research on the Murine Typhus vaccine.

Author response:  As suggested, sentences have been added (lines 417-422) highlighting more on vaccines and vaccine candidates.

  1. Line 447, please elaborate what kind of novel targeted strategies in 1-2 sentences.

Author response:  As suggested, additional sentences have been added (please see lines 455-457) to expand on the statement previously on line 447. 

Reviewer 3 Report

Comments and Suggestions for Authors

Dear Author, thank you for submitting an interesting review entitled „ Murine Typhus: A Review of a Reemerging Flea-Borne Rickettsiosis with Potential for Neurologic Manifestations and Sequalae” because the transfer of information is always useful.

Below you can find some general and specific comments:

General Comments:

The article provides a comprehensive overview of murine typhus, an acute febrile illness caused by Rickettsia typhi. It discusses the disease's etiology, transmission, endemic regions, and its reemergence as a significant cause of febrile illness in certain parts of the United States. Additionally, the article explores the neurologic manifestations of murine typhus and emphasizes the importance of early diagnosis and treatment with doxycycline.

Strengths:

The article effectively presents the key concepts related to murine typhus, its causative agent, reservoirs, vectors, transmission, endemic regions, and the emerging transmission cycle involving opossums and cat fleas in specific areas. The discussion on the neurologic manifestations of the disease adds valuable information.

Weaknesses:

Even if the bibliographic list is exhaustive, most of them are older than 10 years.

I have not been able to identify the link between the title of the paper, which leads me to think that it is a re-emerging disease, and the text, where I do not find much information on episodes of the disease suggesting the re-emergence of the pathogenic entity.

The paper is a good review, which is more like an updated course, rather than material that highlights the re-emerging nature of the disease.

Specific Comments:

Line 10, 11, 16, 17, 26, 30, 50, etc - please edit all species names in italics throughout the text.

Please check throughout the text for spaces between sentences, they seem larger than one space.

Author Response

Reviewer 3:

Author response:  Thank you for your thoughtful review.  The reviewer’s time toward contributing to the peer review process is much appreciated.  I believe the suggestions made, and the resultant changes, greatly improve the manuscript.  I have addressed the reviewer comments below and have included approximate line numbers indicating changes as well as yellow highlighted sections in the revised manuscript. 

Dear Author, thank you for submitting an interesting review entitled „ Murine Typhus: A Review of a Reemerging Flea-Borne Rickettsiosis with Potential for Neurologic Manifestations and Sequalae” because the transfer of information is always useful.

Below you can find some general and specific comments:

General Comments:

The article provides a comprehensive overview of murine typhus, an acute febrile illness caused by Rickettsia typhi. It discusses the disease's etiology, transmission, endemic regions, and its reemergence as a significant cause of febrile illness in certain parts of the United States. Additionally, the article explores the neurologic manifestations of murine typhus and emphasizes the importance of early diagnosis and treatment with doxycycline.

Strengths:

The article effectively presents the key concepts related to murine typhus, its causative agent, reservoirs, vectors, transmission, endemic regions, and the emerging transmission cycle involving opossums and cat fleas in specific areas. The discussion on the neurologic manifestations of the disease adds valuable information.

Weaknesses:

Even if the bibliographic list is exhaustive, most of them are older than 10 years.

Author response:  This is correct.  There is much in the literature that is over 10 years old but still of great value.  I counted the cited references published in the last 10 years (2013 to present) and counted ~79 (representing 47% of the cited references).  I don’t believe this represents a problematic skew toward older references.    

I have not been able to identify the link between the title of the paper, which leads me to think that it is a re-emerging disease, and the text, where I do not find much information on episodes of the disease suggesting the re-emergence of the pathogenic entity.

Author response:  There are several areas within the manuscript that contain potential to emphasize the emerging and reemerging nature of R. typhi infection.  I have tried to place more emphasis on the reemerging nature of R. typhi in this revised version.  To highlight these, and to add more emphasis, I have made several modifications (see response to the next reviewer comment directly below). 

The paper is a good review, which is more like an updated course, rather than material that highlights the re-emerging nature of the disease.

Author response:  The following have been modified or added to highlight the reemerging nature of this infection:

- Lines 152-154 (added a summary sentence that ties in concepts in the preceding sentences to stress the potential for reemergence where rats are present)

- Lines 188-189 (a sentence was added to stress that the cosmopolitan nature of cat fleas can lead to emergence in non-endemic areas)

- Line 190 (changed heading to emphasize discussion on reemergence of this pathogen)

- Line 209 (used a qualifier to emphasize the reemergence of the pathogen in Latin America)

- Lines 213-214 (a sentence was added to bring together concepts from previous sentences that explain the reemerging nature of murine typhus in several regions)

Specific Comments:

Line 10, 11, 16, 17, 26, 30, 50, etc - please edit all species names in italics throughout the text.

Author response:  I apologize for this error.  All genus and species names were placed in italics when submitted.  The original manuscript was submitted simply as a double-spaced Word document.  It seems that the manuscript has since been placed into a templated document (perhaps by editorial staff).  I believe some of the formatting was lost in this transition (at least in the first several paragraphs).  I have re-italicized the genus and species in these sections. 

Please check throughout the text for spaces between sentences, they seem larger than one space.

Author response:  Thank you for noting this. I have the habit of placing two spaces after a sentence (this was the way I was taught to type years ago).  I understand that this old style has been supplanted by use of a single space after a sentence (and adopted by most style guides).  I have gone through the manuscript to place only one space after each sentence.  I believe the manuscript is more visually appealing after this change.  

Round 2

Reviewer 3 Report

Comments and Suggestions for Authors

Dear Authors,

Thank you for the effort you have made to remedy the weaknesses of your paper. The answers and comments you have provided are very satisfying to me.

It is not a big problem to cite old bibliographies, but I have to refer to this because in the MDPI Guidelines for Reviewers (https://www.mdpi.com/reviewers) it is provided (Are the cited references mostly recent publications (within the last 5 years) and relevant? Are any relevant citations omitted? Does it include an excessive number of self-citations?).

I suggest you read the material for reviewers so as not to misinterpret their comments.

Author Response

Thank you for the effort you have made to remedy the weaknesses of your paper. The answers and comments you have provided are very satisfying to me.

Author’s response:  The author thanks the reviewer for their consideration of these revisions.  I believe the changes strengthen the manuscript, and I hope an accepted form will be useful to the journal’s readership.  See below for additional changes made to the manuscript. 

It is not a big problem to cite old bibliographies, but I have to refer to this because in the MDPI Guidelines for Reviewers (https://www.mdpi.com/reviewers) it is provided (Are the cited references mostly recent publications (within the last 5 years) and relevant? Are any relevant citations omitted? Does it include an excessive number of self-citations?).

Author’s response:  I now understand the need for the reviewer to go through these criteria.  To help comply, I have changed a few older references consisting of general material (reviews) rather than original research. I replaced them with some newer references to help bolster the number of citations published within the last 5 years.  These changes include: changing reference 1 (line 472) to include a newer reference published in 2020; changing reference 133 (line 758) to a newer reference published in 2019; and changing reference 148 (line 799) to a newer reference published in 2021. 

I suggest you read the material for reviewers so as not to misinterpret their comments.

Author’s response:  Understandable to have to comment on necessary elements required of the journal.  Thank you for your thoughts and kind review.